# Efficacy of Nurse-Led and Multidisciplinary Self-Management Programmes for Heart Failure with Reduced Ejection Fraction: An Umbrella Systematic Review

**DOI:** 10.3390/biomedicines13081955

**Published:** 2025-08-11

**Authors:** Pupalan Iyngkaran, Taksh Patel, Diana Asadi, Iqra Siddique, Bhawna Gupta, Maximilian de Courten, Fahad Hanna

**Affiliations:** 1Cardiology, University of Notre Dame, Sydney, NSW 2007, Australia; pupalan.iyngkaran@students.torrens.edu.au; 2Program of Public Health, Department of Health and Education, Torrens University Australia, Melbourne, VIC 3000, Australia; bhawna.gupta@torrens.edu.au; 3Health Data Sciences, University of New South Wales, Sydney, NSW 2052, Australia; ttakshp@gmail.com; 4Biomedical Sciences, Monash University, Clayton, VIC 3800, Australia; diana.e.asadi@gmail.com; 5Physiology and Anatomy, University of Melbourne, Parkville, VIC 3010, Australia; iqrasiddique202@gmail.com; 6Institute for Health and Sport (IHES), Victoria University, Melbourne, VIC 8001, Australia; maximilian.decourten@vu.edu.au

**Keywords:** chronic disease self-management, congestive heart failure, mortality, major adverse cardiovascular events, self-management, systematic review, umbrella review

## Abstract

**Background:** Chronic disease self-management (CDSM) programmes are widely recommended for heart failure with reduced ejection fraction (HFrEF), yet evidence on their effectiveness remains mixed. This systematic review synthesises the evidence and critically appraises the findings from multiple systematic reviews on CDSM for congestive heart failure (CHF) with a focus on the impact of nurse-led and multidisciplinary CDSM interventions in adults with HFrEF. **Design:** Systematic review using PRISMA 2020 and AMSTAR-2 guidelines. **Data Sources and Eligibility:** We searched MEDLINE, Embase, CINAHL, Cochrane Library, and other sources for reviews published from 2012 to 2024. Included were systematic reviews of CDSM interventions for adults diagnosed with HFrEF, focusing on mortality, hospital readmissions, quality of life, and self-management behaviours. **Results:** A total of 1050 studies were screened, with 60 studies being counted in the final analysis, including 22 reviews of high quality. Evidence for mortality benefit was limited and inconsistent across reviews. However, moderate-to-high-certainty evidence showed that nurse-led CDSM interventions improved hospital readmission rates and health-related quality of life (HRQoL). Improvements in self-management behaviours such as medication adherence and symptom monitoring were also frequently reported. **Conclusions:** While evidence for a mortality benefit remains inconclusive, this review highlights consistent benefits of nurse-led CDSM interventions in reducing readmissions and improving HRQoL for HFrEF patients. Future research should prioritise standardised outcome reporting, incorporate economic evaluations, and explore patient-centred and culturally tailored approaches to intervention design. PROSPERO registration number CRD42023431539.

## 1. Introduction

Chronic disease self-management (CDSM) programmes, a process by which individuals with long-term health conditions actively engage in managing their symptoms, treatment plans, physical and psychosocial consequences, and lifestyle adjustments [1], have been widely studied across a range of long-term conditions, including congestive heart failure (CHF). Yet, despite this extensive body of research, uncertainty remains regarding which models of CDSM—particularly in terms of delivery and workforce configuration—are most effective for improving outcomes in heart failure with reduced ejection fraction (HFrEF). There is growing recognition that previous systematic reviews, while valuable, often examine heterogeneous interventions or broad disease populations, making it difficult to draw specific conclusions about the comparative efficacy of nurse-led and multidisciplinary CDSM approaches in CHF care.

Decades of research underscore the need to consolidate and re-evaluate the evidence base to inform future patient-centred, outcome-driven strategies [1,2,3,4]. Synthesising existing systematic reviews through an umbrella review provides an opportunity to distil common findings, identify gaps, and clarify what models of CDSM offer the greatest benefit for individuals with HFrEF. This includes considering intervention complexity, adaptability to patient casemix, and implementation challenges in diverse real-world settings [1,2,3,4]. Programmes that are overly intricate—or, conversely, oversimplified—can hinder effective delivery and patient engagement, especially when not tailored to the nuanced needs of CHF populations.

While clinical management of CHF has advanced considerably, especially regarding pharmacological therapies, it remains a complex syndrome requiring sustained, multidisciplinary input and substantial healthcare resources. The public health burden continues to rise due to ageing populations and ongoing challenges in ensuring timely, equitable care—particularly in rural and remote areas. Hospital readmission and mortality rates remain high within 6 to 12 months of discharge, ranging from approximately 25% to 50%, with CHF care consuming 1–2% of healthcare budgets in high-income countries [4,5].

Alongside these clinical developments, non-pharmacological strategies have gained prominence. Landmark studies support the use of structured care models and adherence to performance metrics as a means to enhance uptake of guideline-directed therapy and improve clinical outcomes. However, significant disparities persist—particularly for socioeconomically disadvantaged populations—and health literacy remains a key determinant of self-management capacity. Comorbidities such as coronary artery disease, diabetes, hypertension, renal impairment, and substance use disorders affect over 50% of individuals with CHF, further complicating care delivery and the design of effective CDSM interventions [6].

Despite increasing interest in self-management, questions remain about the effectiveness and sustainability of different CDSM models in CHF care [4,7,8,9,10,11]. This umbrella review aims to synthesise findings from existing systematic reviews to better understand the impact of nurse-led and multidisciplinary CDSM interventions for adults with HFrEF. By focusing on delivery models, workforce roles, and implementation contexts, this review contributes new insights into how integrated, team-based approaches can be optimised to improve patient outcomes.

### Aims

This umbrella systematic review pools systematic reviews for CHF utilising CDSM interventions in the management programme. This study complements an identification of historical gaps [9] and also seeks to identify effective CDSM strategies. Specific research questions are addressed, to gather evidence from systematic reviews on the efficacy of CDSM programme interventions compared to usual care on major adverse cardiovascular outcomes (MACE), HRQoL, and self-management behaviours in a CHF population, focusing on those with heart failure with reduced ejection fraction (HFrEF).

## 2. Methods

This umbrella systematic review followed a pre-published protocol [9,10] registered with PROSPERO (CRD42023431539) and adhered to PRISMA 2020 and AMSTAR 2 Statement [12,13,14,15,16,17,18,19,20,21]. The methodology encompassed five core steps: eligibility criteria, search strategy, data extraction, risk of bias assessment, and data analysis.

### 2.1. Eligibility Criteria

Study Types:

We included systematic reviews and meta-analyses that pooled data from controlled trials examining CDSM interventions in patients with HFrEF. Reviews that relied solely on observational studies were excluded to minimise risk of bias. No restrictions were placed on language or publication date (1980–2024). Grey literature and reference lists were also screened manually, and authors were contacted for unpublished data.

Participants:

Eligible studies enrolled adults (≥18 years) diagnosed with HFrEF (EF < 45%). Participants could have been recruited within 12 months of diagnosis or during admission for acute decompensated heart failure. Studies covering all aetiologies of CHF—such as ischaemic, viral, idiopathic, metabolic, and substance-related causes—were considered. Studies focused solely on HFpEF (EF > 50%) or without a clear description of the CDSM intervention were excluded. No restrictions were applied based on age, gender, or ethnicity.

Interventions:

Eligible interventions involved structured CDSM programmes implemented through a standardised disease management pathway. These included programmes designed to improve MACE and/or self-management behaviours. Health professionals involved ranged from primary care providers and specialists to allied health staff (e.g., nurses, pharmacists, and physiotherapists). Programmes often designated a case manager to coordinate care.

Comparators typically reflected standard guideline-based care in accordance with ACC/AHA and EHA recommendations. Core CDSM components included disease management, self-monitoring, care coordination, coaching, and tailored management strategies.

Types of Outcome Measures:

*Primary Outcomes* concluded on the level and quality of quantitative evidence for CDSM in CHF in reducing MACE. *Secondary Outcomes* include data on self-management behaviours.

### 2.2. Search Strategy

#### 2.2.1. Study Characteristics

Systematic review and meta-analysis of clinical studies were included, and observational studies were limited. However, some pooled studies have a mixture of trial format, hence the need for grading of these reviews and for establishing the quality scale.

#### 2.2.2. Electronic Search

A comprehensive search was conducted in EbscoHost, all databases, Medline, PubMed (1950–2022), the Cochrane Register of Controlled trials (CENTRAL) (2023), Embase (1980 to 2023), CINAHL (1982–2023), PsycINFO (1887–2023), Science Citation Index (1987–2023), and Web of Science. The MeSH terms used to shortlist studies were previously published [8,9]. In addition, a ‘snowball’ search of relevant selected reviews was conducted.

#### 2.2.3. Searching Other Resources and Information Sources

This was conducted as the study evolved and as required. All new changes are documented as an amendment to the protocol.

### 2.3. Data Collection and Analysis

#### 2.3.1. Study Selection

Title and abstract screening was conducted independently by two reviewers (P.I. and T.P.) to assess eligibility based on predefined inclusion criteria. Full texts of potentially relevant studies were then reviewed in detail, also independently and unblinded, by the same reviewers. Discrepancies during the selection process were discussed and resolved by consensus. A third reviewer (F.H.) acted as arbiter and reviewed all stages of the process, resolving any unresolved disagreements or uncertainties.

In some cases, study authors were contacted to clarify missing or unclear results. Where clarification could not be obtained, studies were excluded from the main analysis.

#### 2.3.2. Data Extraction and Management

Data extraction was performed independently by two reviewers using a structured template developed specifically for this review. Key data fields included the following:Study quality and risk of bias;Review characteristics (e.g., number and type of included studies);Population and setting;CDSM intervention characteristics;Outcomes and findings relevant to CHF.

When results for a single study were reported across multiple publications, data were consolidated to ensure comprehensive representation.

#### 2.3.3. Data Items

The data extraction template incorporated CHF-specific domains and outcome measures based on previous umbrella reviews and disease-specific guidance [6,7,8,9]. Domains included patient adherence, hospitalisation rates, mortality, quality of life, and implementation-related factors.

#### 2.3.4. Quality Appraisal and Risk of Bias

Two reviewers (F.F. and T.P.) independently assessed the methodological quality of the included systematic reviews. The following tools were used:AMSTAR 2 for assessing the quality of systematic reviews [12];RoB 2.0, the revised Cochrane risk of bias tool, for included RCTs [14,15].

Discrepancies were resolved through discussion or consultation with a third reviewer.

#### 2.3.5. Certainty of Evidence

The certainty of evidence for key outcomes was evaluated using the GRADE (Grading of Recommendations Assessment, Development, and Evaluation) framework. This assessment considered five domains:Risk of bias;Inconsistency;Indirectness;Imprecision;Publication bias.

Evidence was then categorised as very low, low, moderate, or high certainty [21].

*Reporting of excluded studies* is provided in the PRISMA figure. Information is also provided in the Results and Appendix A.

#### 2.3.6. Data Analysis, Assessment of Heterogeneity, and Publication Bias


*Planned methods for study analysis and statistical methodology*


The searching process and data extraction was guided by Cochrane Handbook for Systematic Reviews of Interventions [15]. The initial step was summary of included findings in a table and qualitative analysis. Subgroup analyses were performed on key domains, e.g., various comorbid conditions, training of intervention arms, treatment intensity, and other unanticipated factors [10,11,12,13,14,15,16,17,18,19,20,21,22].


*Risk of bias in individual studies*


We used the Cochrane Risk of Bias tool, with two reviewers, in pairs, to assess and rate risk of bias from standards defined in the Cochrane Collaboration’s tool [13,14,15,16,17].

### 2.4. Measures of Treatment Effect

The results are presented as a risk ratio with a 95% confidence interval (CI) to express estimates of effects for dichotomous variables and outcomes. For continuous variables, results are expressed as the mean difference with 95% CI. For outcomes measured using a variety of methods, the size of the intervention effect is presented as standardised mean difference with 95% CI.

### 2.5. Ethics and Dissemination

Ethical approval is not required for the study as no primary patient data are collected. This review will extract current and comprehensive research publications on CDSM and CHF. At this juncture, it is vital to inform the literature on the efficacy of CDSM within the CHF context. It is important to plan studies to counter the downgrade of evidence and inform future guidelines.

## 3. Results

Search Results

The summary of the search is highlighted in PRISMA chart (Figure 1). From a total 974 identified potentially eligible systematic reviews, 58 reports from 57 studies were included (Table 1 and Table 2). Studies were divided into three categories: case management (*n* = 10), nurse-led interventions (*n* = 8), and non-pharmacological interventions (*n* = 41). The total number of studies reported for each study is 1430 (range 6–105 articles); the cumulative population from each study is 276,381 (range 467 to 76,582 patients). This value does not factor overlap RCT’s in various SRs.

The quality appraisal of studies was conducted as per the systematic protocol, using AMSTAR-2 grading [21,23,24,25,26,27,28,29,30,31,32,33,34,35,36,37,38,39,40,41,42,43,44,45,46,47,48,49,50,51,52,53,54,55,56,57,58,59,60,61,62,63,64,65,66,67,68,69,70,71,72,73,74,75,76,77,78,79,80,81,82,83,84]. The mean quality assessment (QA) score was low to moderate 40/61 (66%). The study quality rated 22 high, 21 moderate, 17 low, and 1 very low. Several studies that had a partial yes response were included as high after a secondary review, and gaps were deemed not significant [28,31,32,50,67,68,73,74,81]. AMSTAR-2 questions 2, 4, 7, and 8 on review methods and deviation of methodology, search strategy, list of excluded studies and justifications, description of study details scored the worst where >50% studies were partial yes or no. Meta-analysis was not performed in 14 studies [23,38,41,47,58,64,65,70,71,75,81]. There were fifteen non-randomised trials [38,41,47,50,53,54,55,56,57,58,59,60,64,65,71,77] included in this study. There was no discernible trend in review quality across intervention categories across high-quality studies, and it declined from moderate to very low studies. Scoping data from 2 studies [25,72] informed studies in tables with no own data included in the actual tables. The quality of and type of intervention varied across the studies in Table 3, the high-quality studies in Table 4 and excluded students Table A1.


**AMSTAR Questions**


Did the research questions and inclusion criteria for the review include the components of PICO?Did the report of the review contain an explicit statement that the review methods were established prior to the conduct of the review, and did the report justify any significant deviations from the protocol?Did the review authors explain their selection of the study designs for inclusion in the review?Did the review authors use a comprehensive literature search strategy?Did the review authors perform study selection in duplicate?Did the review authors perform data extraction in duplicate?Did the review authors provide a list of excluded studies and justify the exclusions?Did the review authors describe the included studies in adequate detail?Did the review authors use a satisfactory technique for assessing the risk of bias (RoB) in individual studies that were included in the review? (RCTs; NRSI)Did the review authors report on the sources of funding for the studies included in the review?If meta-analysis was performed, did the review authors use appropriate methods for statistical combination of results? (RCTs; NRSI)If meta-analysis was performed, did the review authors assess the potential impact of RoB in individual studies on the results of the meta-analysis or other evidence synthesis?Did the review authors account for RoB in individual studies when interpreting/discussing the results of the review?Did the review authors provide a satisfactory explanation for, and discussion of, any heterogeneity observed in the results of the review?If they performed quantitative synthesis, did the review authors carry out an adequate investigation of publication bias (small study bias) and discuss its likely impact on the results of the review?Did the review authors report any potential sources of conflict of interest, including any funding they received for conducting the review?


**Table 2 AMSTAR Scoring and interpretation**


Rating overall confidence in the results of the review

High—Zero or one non-critical weakness: The systematic review provides an accurate and comprehensive summary of the results of the available studies that address the question of interest.Moderate—More than one non-critical weakness: The systematic review has more than one weakness but no critical flaws. It may provide an accurate summary of the results of the available studies that were included in the review.Low—One critical flaw with or without non-critical weaknesses: The review has a critical flaw and may not provide an accurate and comprehensive summary of the available studies that address the question of interest.Critically low—More than one critical flaw with or without non-critical weaknesses: The review has more than one critical flaw and should not be relied on to provide an accurate and comprehensive summary of the available studies.Note: Multiple non-critical weaknesses may diminish confidence in the review, and it may be appropriate to move the overall appraisal down from moderate to low confidence.

https://amstar.ca/Amstar-2.php (accessed on 22 January 2025) Adapted from Table 4 [10].

i.Effects by cardiovascular outcomes
a.Mortality

Mortality outcomes were reported in 31 studies across the included reviews. Of these,

12 studies reported a significant reduction in mortality associated with CDSM interventions, particularly those delivered through nurse-led or team-based models [23,24,26,31,32,39,43,46,57,67,68,80].Three studies indicated a positive trend toward mortality reduction, though not statistically significant [63,66,79].Six studies presented equivocal findings, with mixed or inconclusive mortality outcomes [30,33,34,49,64,76].10 studies found no mortality benefit or reported a negative effect, including either no change or increased mortality rates [27,28,35,44,47,48,49,62,81,82].28 studies did not report mortality outcomes [25,29,36,37,38,40,41,42,45,50,51,52,53,54,55,56,58,59,60,61,65,69,70,72,74,75,77,78].

By Study Quality:

High-Quality Reviews

Significant mortality reduction: [31,32,67,68,71,80].Equivocal: [30,49,66,79].Negative: [27,35,62,81,82].Not reported: [42,50,52,72,74,77,78].

Moderate-Quality Reviews

Significant mortality reduction: [43].Equivocal: [33,76].Negative: [47,48,73,75].Not reported: [36,38,41,51,53,54,56,58,59,65].

Summary Insight

Evidence for the impact of CDSM on mortality was mixed. A minority of high-quality studies showed significant mortality reduction, particularly in structured nurse-led or integrated multidisciplinary models. However, a substantial number of studies either reported no mortality benefit or did not assess this outcome at all. The inconsistency may reflect variations in study design, follow-up duration, or intervention intensity. As mortality was not a primary outcome in many included reviews, these findings should be interpreted with caution.

ii.Hospital readmissions

Hospital readmission data were reported in 34 studies across the umbrella review. Of these,

25 studies reported a significant reduction in readmissions following CDSM interventions. These effects were most commonly associated with structured, nurse-led models [23,24,27,29,31,32,35,39,41,43,46,47,52,56,57,58,59,62,66,67,68,71,73,74,81,82].Three studies showed a positive trend, though it was not statistically significant [26,63,79].Seven studies were equivocal, showing mixed or inconclusive results [30,33,34,37,49,64,76].Three studies reported no improvement or a negative effect on readmissions [28,44,48].21 studies did not report readmission outcomes [25,33,38,42,45,50,51,53,54,55,60,61,65,69,70,72,75,77,78,80].

By Study Quality:

High-Quality Reviews

Significant reduction in readmissions: [27,31,32,35,62,66,67,68,71,74,81,82].Equivocal findings: [30,49,79].Negative findings: None reported.Not reported: [42,50,67,77,78,80].

Moderate-Quality Reviews

Significant reduction in readmissions: [39,41,43,46,47,56,58,59,73].Equivocal: [33,76].Negative: [48].Not reported: [36,38,45,51,53,54,65,75].

Summary Insight

Evidence for the impact of CDSM interventions on hospital readmissions was generally favourable. A substantial number of both high- and moderate-quality reviews reported significant reductions, particularly for nurse-led and structured multidisciplinary programmes. While a small number of studies found no effect or had equivocal results, no high-quality studies reported a significant worsening of readmission outcomes. Variability in follow-up duration and outcome reporting, however, limits the strength of causal claims in some cases.


b.Health-related quality of life


HRQoL outcomes were reported across the 29 studies included in the umbrella review. Of these,

23 studies showed statistically significant improvements in HRQoL following CDSM interventions, particularly those led by nurses or multidisciplinary teams [23,24,25,29,33,34,35,37,38,43,44,45,46,47,50,57,58,59,65,66,67,68,69,70,71,73,74,80].Seven studies reported a positive trend toward HRQoL improvement, though it was not always statistically significant [32,48,61,62,63,79,82].Nine studies were equivocal, with mixed or inconclusive findings [31,40,42,49,56,60,64,76,81].Four studies found no improvement or a decline in HRQoL scores [29,40,52,61].14 studies did not report on HRQoL at all [26,27,30,36,41,52,53,54,55,72,75,77,78].

Improvements were most consistent in nurse-led interventions, especially those involving structured education, regular follow-up, and patient goal-setting. Reviews highlighted the importance of culturally and contextually adapted delivery models in optimising patient-reported quality of life outcomes.


c.Self-management behaviours


Self-management (SM) outcomes were reported across 31 studies in the umbrella review. Of these,

25 studies showed statistically significant improvements in SM behaviours, including medication adherence, symptom monitoring, dietary/lifestyle modifications, and patient confidence [23,25,26,28,29,36,38,42,43,44,46,47,49,50,51,53,57,58,59,60,67,69,70,71,73,74,75,76,77,78,80].Four studies reported a positive trend toward SM improvement, though it was not statistically significant [62,64,67,79].10 studies had equivocal findings, with mixed or inconclusive evidence [34,36,39,40,53,54,55,63,67,81].Four studies found no improvement or a decline in SM behaviours [39,45,48,61].Nine studies did not report on self-management outcomes [24,27,30,31,32,33,34,52,66].

By Study Quality:

High-Quality Reviews

Significant SM improvement reported: [42,49,50,67,68,71,74,77,78,80].Equivocal findings: [35,68,72,79,81,82].Negative outcome: [62].Not reported: [27,30,31,32,52,66].

Moderate-Quality Reviews

Significant SM improvement reported: [33,36,38,43,46,47,51,53,58,59,73,75,76].Equivocal findings: [41,54,56,65].Negative outcome: [39,45,48].Not reported: [33].

Summary Insight

Overall, the studies led by nurses or delivered through structured, multidisciplinary approaches with patient education and follow-up performed best. Improvements were particularly robust in high-quality reviews, though some heterogeneity in outcomes remains—likely due to differences in intervention intensity, population characteristics, and outcome measures used.

iii.Effects by intervention on health services

Cost-effectiveness of studies was not well documented or not reported in most studies. Three areas including the utility of emergency services including emergency rooms showed positive benefits in [26,57], trends in [63,64], and no benefit [28,29]; the length of hospital stay as beneficial [28,56,57,66,71], equivocal [63,64], or with no benefits [30] and the cost of care were explored also as beneficial [28,29,35,46,52,56,57,71,74,80,81,82], negative [62], or equivocal [63,64]. Only these studies were of moderate-to-high quality [33,35,46,52,56,62,66,71,74,80,81,82]. The details of the studies, care domains, lead providers, and self-management domains are reported below.

iv.Results synthesis and summary

Across the umbrella review, consistent patterns emerged regarding the impact of nurse-led and multidisciplinary CDSM interventions on key outcomes in HFrEF care. Improvements in HRQoL and self-management behaviours were the most consistently reported benefits, with the strongest effects observed in nurse-led models, particularly those involving structured education, regular follow-up, and tailored patient engagement. These findings were supported by both high- and moderate-quality reviews.

Secondary outcomes such as hospital readmissions and mortality were less frequently reported and demonstrated more variability, with trends suggesting potential benefit but limited conclusive evidence. Multidisciplinary interventions that included pharmacists, psychologists, or allied health professionals showed added value, particularly in addressing comorbidities and social determinants of health.

Table 5, Table 6 and Table 7 provide a consolidation of these findings by outcome domain, allowing clearer comparison across studies and quality tiers. Overall, the evidence supports a growing consensus around the effectiveness of integrated, nurse-led, or team-based CDSM approaches in improving patient-reported outcomes in HFrEF populations.

## 4. Discussion

Summary of Evidence

This umbrella review synthesises four decades of systematic reviews on chronic disease self-management (CDSM) in heart failure with reduced ejection fraction (HFrEF). While CDSM has long been considered a foundational strategy in chronic disease care, recent revisions to the 2022 ACC/AHA guidelines have downgraded it from a performance to a quality measure. This reflects evolving priorities in guideline development and growing emphasis on health-related quality of life (HRQoL) rather than major adverse cardiovascular events (MACEs) as the primary outcome of interest.

The findings from this review reaffirm that well-structured, nurse-led, and multidisciplinary CDSM interventions can improve patient outcomes when implemented with fidelity and contextual relevance. However, variability in study quality, design, delivery models, and reporting standards remains a key challenge, particularly in comparing outcomes across clinical settings and populations [83,84,85,86,87,88,89].

Mortality

While some reviews report mortality benefits associated with CDSM, the overall picture remains mixed. Differences in intervention intensity, care models, and follow-up periods likely contribute to inconsistent findings. Our review reflects similar variability reported in other chronic conditions. For instance, a recent CDSM study in COPD populations also failed to demonstrate consistent mortality reduction, despite improvements in other outcomes [88]. These findings suggest that while mortality is an important endpoint, its sensitivity to complex, non-pharmacological interventions may be limited—particularly when therapeutic environments and standard care are already optimised.

Hospital Readmissions

Hospital readmissions were more consistently impacted by CDSM interventions, particularly those involving structured follow-up and multidisciplinary collaboration. These findings are consistent with other chronic disease settings, where CDSM training and patient support strategies have reduced healthcare utilisation [70,88]. Standardising how readmissions are reported and tracked will further support the integration of these interventions into care models and funding frameworks.

Health-Related Quality of Life (HRQoL)

HRQoL improvements emerged as one of the most consistent benefits across reviewed studies. Interventions that included emotional, educational, and behavioural support were particularly associated with better quality of life. These findings align with the broader literature, which shows that HRQoL can be effectively enhanced through patient-centred self-management approaches [88,89]. Given the multidimensional nature of HRQoL, interventions must be adaptable to patient preferences, cultural norms, and health literacy levels to optimise impact.

Self-Management Behaviours

CDSM interventions demonstrated strong potential to enhance patient engagement, medication adherence, and proactive health behaviours. This is further supported by external literature; for example, a systematic review and meta-analysis by Cho and Kim found that nurse-led self-management programmes improved behavioural outcomes and significantly reduced HbA1c in people living with diabetes [89]. These findings highlight the cross-condition benefits of nurse-led education and support, which are central to empowering patients and reducing dependency on acute services.

Health Service Effects

While evidence on service use and cost-effectiveness was less frequently reported, the available data point to potential reductions in emergency visits and shorter hospital stays. However, without consistent reporting and economic modelling, conclusions remain tentative. Given the rising burden on healthcare systems, particularly in the context of multimorbidity and an ageing population, future evaluations must include resource utilisation and cost data to guide scale-up and investment decisions.

Making Sense of High-Quality Evidence

Earlier high-quality reviews—such as those by Roccafort, Ruppar, Jonkman, Boren, and Inglis—provided strong evidence of benefit across outcomes including mortality, self-management, and hospitalisation. These reviews consistently included structured, multidisciplinary, and nurse-led programmes with extended follow-up durations and involvement of family carers. Importantly, these studies were conducted before widespread uptake of modern pharmacotherapies, suggesting that CDSM interventions held value even in the absence of today’s advanced therapies.

In contrast, more recent high-quality reviews such as those by Lee [48], Ditewig, and Paranjuli [79] offered more equivocal or negative conclusions. These differences may reflect changes in comparator care (e.g., higher baseline quality), variability in delivery agents (e.g., pharmacists vs. nurses), or inconsistency in intervention scope. Nonetheless, these findings reinforce that the effectiveness of CDSM is not solely about content but also who delivers it, how, and in what context.


**Strengths and Limitations**


This umbrella review is the first to synthesise systematic reviews specifically focused on nurse-led and multidisciplinary CDSM in HFrEF. A key strength lies in its comprehensive scope, inclusion of high-quality evidence, and alignment with chronic disease management frameworks.

However, limitations include the inability to perform meta-analysis and the reliance on secondary interpretations of primary study data. The heterogeneity in intervention formats, delivery personnel, and outcome definitions across reviews limits the precision with which definitive conclusions can be drawn. Additionally, many reviews lacked detailed reporting on economic outcomes, cultural adaptations, or implementation factors—key elements for translating findings into policy and practice.


**Implications for Practice and Policy**


Nurse-led CDSM should be embedded into routine HFrEF care as part of standard, multidisciplinary disease management programmes.Future trials should adopt a core outcomes set that includes HRQoL, hospital readmissions, self-management behaviours, and cost-effectiveness.Co-design and cultural tailoring should be prioritised to enhance relevance and patient engagement across diverse settings.Economic evaluations must be incorporated into programme design to support scale-up and policy adoption.

## 5. Conclusions

This umbrella review highlights consistent benefits of nurse-led and multidisciplinary CDSM interventions for individuals with HFrEF, particularly in reducing hospital readmissions and improving HRQoL. While evidence for mortality reduction remains mixed—even among high-quality reviews, these interventions appear to play a critical role in stabilising patient trajectories and enhancing everyday functioning.

Persistent gaps include underreporting of cost-effectiveness, limited analysis of culturally tailored models, and variability in outcome measurement. Addressing these gaps through more standardised, equity-oriented, and economically evaluated research will be essential to support broader implementation and inform future policy decisions.

## Figures and Tables

**Figure 1 biomedicines-13-01955-f001:**
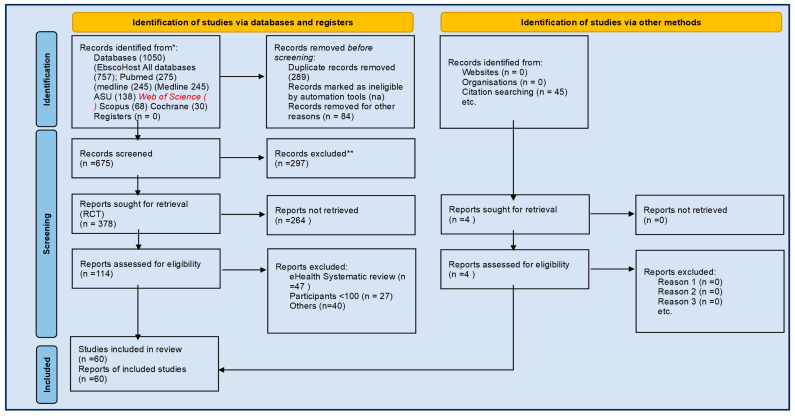
PRISMA 2020 flow diagram for new systematic reviews, which included searches of databases, registers, and other sources. * Search previously detailed from Refs. [9,10].

**Table 1 biomedicines-13-01955-t001:** Summary (characteristics) of included studies.

Author (Year); Country	Study Details				Study Intervention	Summary of Intervention
	Type/Number	Participants	F/U	Database		
Zhao et al., 2024 China [23]	RCT 20	P = 4681 R = 10–1518	NA	Em, Md, Pb, SD, WS [In to 2022]	Case Management	4 different MDT care models
Chen et al., 2023 Taiwan [24]	RCT + MA 13	P = 2666 R = 28–197	2–24 m	CI, CL, Pb, Md [2002–2022]	Case Management	Collaborative health Management
Li 2023 China [25]	RCT + MA 10	P = NA R = NA	3–12 m	CE, CK, Eb, Pb, WoS [NA]	Case Management	Transitional care
Yang Mly 2023 [26]	RCT + MA 105	P = 37,607 R = NA	6–12 m	CL, Em, Ov, Pb [In–2022]	Case Management	Multicomponent integrated care
Hafkamp 2022 Holland [27]	RCT + MA 44/186	P = 6101 R = 40–1650	3–34 m	CI, CL, Pb, PsI WoS [2011–2021]	Case Management	Care pathways
Hsu 2022 Taiwan [28]	RCT-Co + MA 6/1	N = 2346 R = 40–1937	NA	CI, CL, CP, Em, Pb [2020]	Case Management	Patient navigators
Toback 2017 Canada [29]	RCT-NRT 26	NA	NA	Pb, UP [1999–2016]	Case Management	Multiple self-management support
Taylor 2005 UK [30]	RCT 21	P = 1627	NA	Am, D, CI, CL, Em, Med; NHS, NRR, SCI; Si [2003]	Case Management	Disease management (MDT, CMM, CM)
Roccaforte 2005 [31]	RCT + MA 33	P = 7538 R = 34 to 1518	3–22 m	CL; Em; Med; Pb [1980–2004]	Case Management	Disease management programmes
Gonseth 2004 [32]	RCT-Co + MA 54	NA R = 34–1966	1–50.4 m	CL, Em, Med [1966–2003]	Case Management	Disease management programmes—elderly
Huang 2023 [33]	RCT + MA 25	P = 2746 R 40 to 228	1 w–12 m	CE, CI, Em, Med, PsI, WoS (In to 2022)	Nurse-Led	Self-management
Nwosu 2023 [34]	SR 18	P = 2413	NA	CI, Med, PsI, WoS (In to 2022)	Nurse-Led	Patient education
Checa 2022 [35]	RCT-QE-Co + MA 30	P = 8209 R-24 to 1894	30 d–12 m	CE, CL, CT, Em, ICTRP, Med, WHO (In to 2022)	Nurse-Led	Case management primary care
Huang 2022 [36]	RCT + MA 24	P = 2488 R-36 to 382	30 d-12 m	CE, CI, Em, Med, PsI, WoS (In to 2021)	Nurse-Led	Self-management
Ceu 2022 [37]	RCT-NRT 9	NA	NA	CI, Med	Nurse-Led	Variable nursing interventions
Imanuel Tonapa 2022 [38]	RCT + MA 12	P = 1938 R 36–1437	2 m–12 m	CI, CL, Em, Med, Ov, Pb, WoS (In to 2020)	Nurse-Led	Telecoaching
Son 2020 [39]	RCT + MA 8	P = 1979 R 88–412	3 m–12 m	CI, CL, Em, Pb, WoS (2000 to 2019)	Nurse-Led	Self-management education
Walsh 2017 Abs Conf [40]	RCT-NRT 68	NA	NA	CI, Pb, Med, SC (2006 to 2016)	Nurse-Led	Clinic-based self-management education
Alnomasy 2023 [41]	RCT + MA 14	P = 2035 R: 40–767	30 d–12 m	CL; NAHL; Pb; WoS (NA)	Non-Pharmacological	Ambulatory—home visits, phone calls, digital platforms, technologies
Mhanna 2023 [42]	RCT + MA 6	P = 489 R: 41–158	30 d-12 m	CL; Em; Med; Pb (In to 2022)	Non-Pharmacological	CBT
Olano-Lizarraga 2023 [43]	RCT 8	P = 1623 R: 64–468	30 d-12 m	CI; CL; Pb; PsI; SC; (2010 to 2022)	Non-Pharmacological	Interventions targeting the social dimension
Nso 2023 [44]	RCT + MA 9	P = 1070	3–6 m	Pb; Sco (NA)	Non-Pharmacological	CBT
Balata 2023 [45]	RCT + MA 7	P = 611 R: 26–158	4–32 wk	CL, Pb, SC, WoS (In to 2022)	Non-Pharmacological	CBT
Koikai 2023 [46]	RCT-NCT N = 30	P = 7685 R: 50–1223	NA	CL, Em, GS Pb, SD (2012 to 2022)	Non-Pharmacological	Self-management education strategies
Feng 2023 [47]	RCT + MA 20	P = 3459 R: 39–317	3–12 m	CK, Pb, WoS, VIP (1999 to 2022)	Non-Pharmacological	Self-management intervention strategies
Nahlen Bose 2023 [48]	Meta-review SR = 7, RCT = 67	P = 10,132 R = 320 to 3837	NA	CI, CL Pb, PsI (NA)	Non-Pharmacological	Psychosocial interventions
Lee/Reigel 2022 USA [49]	RCT + MA 27	P = 6950 R = NA	NA	CI, Em, Pb, PsI (2008–2019)	Non-Pharmacological	Self-management intervention
Villero-Jimenez 2022 Spain [50] L-SP-ENG	RCT-NCT 12	P = 1380 R = 19–369	NA	CI, Pb, PsI (NA)	Non-Pharmacological	Dyadic self-management interventions
Ghizzardi 2022 [51]	RCT + MA 9	P = 1214 R- 30 to 510	1–16 m	CI, Em, Pb PsI, SC (In to 2020)	Non-Pharmacological	Motivational interviewing on self-management
Suksatan 2022 Thailand [52]	RCT + NCT 15	P = 10,701 R-36–2494	30 d	CI, CL, Pb, PsI, SC (2011 to 2022)	Non-Pharmacological	Transitional care intervention—elderly
Meng 2021 China [53]	RCT + MA 8	P = 1707 R: 20–902	6 m-42 m	CL, CNKI, Em, Pb (2000 to 2020)	Non-Pharmacological	SM intervention in Knowledge, Attitude, Practice
Tinoco 2021 Brazil [54]	RCT-NRT + MA 19	N = 1841	NA	CI, LI, Pb, SC (2012 to 2019)	Non-Pharmacological	Health education and self-management
Aghajanloo 2021 Iran [55]	RCT-NRT + MA 39	P = 8958 R: 17–2082	NA	Em, GS, Ma, Pb, SID, WoS, (2004 to 2018)	Non-Pharmacological	Self-management behaviours with SCHFI
Cañon-Montañez 2021 Colombia [56]	RCT + RCT 45	P = 9688 R: 37–1049	3–18 m	CI, CL, Em, Li, Pb, SC, WoS (In to 2019)	Non-Pharmacological	Educational Intervention
Anderson 2021 UK [57]	RCT + QE 12	P = 3887 R: 25–1023	3–24 m	BNI, CI, Em, Med (2008 to 2020)	Non-Pharmacological	Advanced-level nurses specialist nurses vs. physician led
Zhao 2021 China [58]	RCT + MA 15	P = 2630 R: 28–475	2 w-12 m	CL, Em, Pb, WoS (In to 2019)	Non-Pharmacological	Self-management interventions
Poudel 2020 USA [59]	RCT + NRT 8	P = 758 R: 30–241	NA	CI, CL, GS, HS, Med, PsI (1990 to 2019)	Non-Pharmacological	Motivational interviewing
Świątoniowska-Lonc 2020 [60]	RCT + MA 16	P = 944 60–1160	1–18 m	Med, Pb, SC (2010 to 2019)	Non-Pharmacological	Health education
Peng 2019 [61]	RCT + MA 8	P = 480 R: 17–158	NA	CL, Em, Pb (In to 2018)	Non-Pharmacological	CBT
Parajuli 2019 Australia [62]	RCT + MA 18	P = 4630 R: 34 to 2169	6 w-55 m	CI, CL, Em, Med, Pb, SC, WoS (In to 2017)	Non-Pharmacological	Pharmacist-involved MDT
Shanbhag 2018 Canada [63]	RCT-NRT 38	P = 76,582 R = 68–50,678	NA	CI, CL, Em, Med (1990–2017)	Non-Pharmacological	Interventions improving physician adherence to guideline
Sterling 2018 USA [64]	RCT-NRT N = 6	P = NA R-NA	NA	AgeLine, CI, CL, Em, Med (In-2017)	Non-Pharmacological	Home care workers
Jiang 2018 Taiwan [65]	RCT + MA 29	P = 3837	1–48 m	CI, CL, Em, Pb, PsI, SC, WoS, ProQuest Dissertation (16 February 2006)	Non-Pharmacological	Psychological interventions on self-management
Jonkman 2016 Holland [66]	RCT + MA 20	P = 5624 R: 42–1023	3–18	CI, CL, Em, Pb, PsI (1985–2013)	Non-Pharmacological	Self-management and programme characteristics
Ruppar 2016 USA [67]	RCT-NCT + MA N = 57	P = 4527 R:10–1518	NA	CI, CL, D, IPA, Highwire Med, SC, PQ (In–2013)	Non-Pharmacological	Medication adherence interventions
Jonkman 2016 Holland [68]	RCT + RCT 20	P = 5624 R: 42–1023	3–18	CI, CL, Em, Pb, PsI (1985–2013)	Non-Pharmacological	Self-management interventions
Srisuk 2016 Thailand [69]	RCT 9	P = 666 R: 61–155	NA	CI, CL, Em, Med, Pb, PsI, SC, WoS (2005–2015)	Non-Pharmacological	Family-based education
Ha Dinh 2016 Vietnam [70]	RCT + NCT 12 (2 HF)	P = 467 (3 HF) R: 88–276	12–15	CI, CL, Em Med, WoS (In–2013)	Non-Pharmacological	Teach-back method and self-management
Inglis 2015 Australia [71]	RCT + MA 41	P = 9332	38% < 6 m	CENRAL, DARE, HTA, Med, Em, CI, SCI, AMED (In–2015)	Non-Pharmacological	Structured telephone support, non-invasive telemonitoring
Ruppar 2015 [72]	RCT + NRT 29	P = 4285	11 d-24 m	CI, CL, Em, Med (In–2013)	Non-Pharmacological	Medication adherence interventions
Casimir 2014 [73]	RCT 7	P: NA R: 121–314	NA	CINAHL, Pb, PsychINFO, EMBASE, CENTRAL, ERIC, SC, DynaMed. (1990–2013)	Non-Pharmacological	Patient centred self-management
Wakefield 2013 USA [74]	RCT + MA 43	P = 8071 R: 48–1518	10 d-549 d	CI, CL, Med (1995–2008)	Non-Pharmacological	Care management programme
Barnason 2012 USA [75]	IR-RCT 19	P:NA R: 18–902	NA	CI, CL, Med, PsI (2000–2010)	Non-Pharmacological	Self-management interventions
Boyde 2011 USA [76]	RCT 19	P = 2686 R: 36–314	3 m-288 d	CI, CL, Em, Med, PsI (1998–2008)	Non-Pharmacological	Educational interventions
Dickson 2011 USA [77]	RCT + MA N = 3	P = 99 NA	NA	NA	Non-Pharmacological	Self-care practices
Yehle 2010 USA [78]	RCT + NRT 12	P = 1360 R: 20–151	NA	Pb, CL, CINAHL, Med, ERIC, Academic Search Premier, Health Source (1966–2009)	Non-Pharmacological	Educational interventions
Ditewig 2010 Holland [79]	RCT 19	P = 4011 R: 50–766	6–24 m	CI, CL, Em, Med (1996–2009)	Non-Pharmacological	Self-management interventions
Boren 2009 USA [80]	RCT 35	P = 7413 R: 36–713	3–18 m	CI, CL, Med (1966–2007)	Non-Pharmacological	Self-management education
Jovicic 2006 Canada [81]	RCT 6	P = 857 R: 70–223	3–12 m	ACP, CI, CL, Em, Med (1966–2005)	Non-Pharmacological	Self-management intervention
McAlister 2004 Canada [82]	RCT 29	P = 5039 R: 34–1396	1–12	AMED, CI, CL, Em, Med (1966–2003)	Non-Pharmacological	Multidisciplinary strategies

Abbreviation: Databases: Am—AMED; BNI—British Nursing Index; CE—CENTRAL; CI—CINAHL; CK—CNKI; CL—Cochrane Library; CP—Chinese Electronic Periodical Services; D—DARE; Eb—EBSCO; Em—Embase; GS—Google Scholar; HS—Health Source; ICTRP—Registry of International Clinical Trials; IPA—International Pharmaceutical Abstracts; Li—LILACS; Ma—Magiran; Med—Medline; NRR—National Research Register (July 2003); NAHL—Nursing and Allied Health Literature; NHS—National Health Services; Ov—Ovid; Pb—PubMed; PQ—ProQuest; Ps—PsycInfo; S—Science Direct; SC—Scopus; SCI—Science Citation Index; Si—SIGLE; SID—Scientific Information Database; Up-To-Date; WHO—World Health Organisation; VIP—Wan Fang, Wei Pu; WoS—Web of Science.Terminology; MA—metaanalysis; MDT—multidisciplinary team; NA—not available; NRT—non roandomised trial; RCT—randomised controlled trial; RCT-QE-Co—quasiexperimental trial; SD—standard deviation; SR—systematic review; Adapted from Refs. [9,10].

**Table 2 biomedicines-13-01955-t002:** Rating (classification) of intervention effectiveness for each outcome and intervention category.

Author (Year); Country	Death/HFM	Readmission	Quality of Life (Depression/Anxiety)	Self-Management Behaviour Ability	A&E Use	Length of Stay	Cost	Strength of Evidence
Zhao et al., 2024 China [23]	+	+	+	+ +	NA	NA	NA	L
Chen et al., 2023 Taiwan [24]	+	+ +	+	NA	NA	NA	NA	L
Li 2023 China [25]	NA	NA	+ +	+	NA	NA	NA	L
Yang Mly 2023 [26]	+	+ −	NA	+	+	NA	NA	L
Hafkamp 2022 Holland [27]	−	+ +	NA	NA	NA	NA	NA	H
Hsu 2022 Taiwan [28]	−	−	−	+	−	+	+	L
Toback 2017 Canada [29]	NA	+	+	+	−	−	+	VL
Taylor 2005 UK [30]	~	~	NA	NA	NA	NA	NA	H
Roccaforte 2005 [31]	+	+ +	~	NA	NA	NA	NA	H
Gonseth 2004 [32]	+	+ +	~ +	~+	NA	NA	NA	H
Huang 2023 [33]	~	~	+	NA	~	NA	NA	M
Nwosu 2023 [34]	~	~	+	NA	NA	NA	NA	L
Checa 2022 [35]	−	+ +	+	~	NA	NA	+	H
Huang 2022 [36]	NA	NA	NA	+	NA	NA	NA	M
Ceu 2022 [37]	NA	~	+	~	NA	NA	NA	VL
Imanuel Tonapa 2022 [38]	NA	NA	+	+	NA	NA	NA	M
Son 2020 [39]	+	+	−	-	NA	NA	NA	M
Walsh 2017 Abs Conf [40]	NA	NA	~	~	NA	NA	NA	L
Alnomasy 2023 [41]	NA	+	NA	~	NA	NA	NA	M
Mhanna 2023 [42]	NA	NA	~	+	NA	NA	NA	H
Olano-Lizarraga 2023 [43]	+	+	+	+	NA	NA	NA	M
Nso 2023 [44]	−	−	+	+	NA	NA	NA	L
Balata 2023 [45]	NA	NA	+	−	NA	NA	NA	M
Koikai 2023 [46]	+	+	+	+	NA	NA	+	M
Feng 2023 [47]	−	+	+	+	NA	NA	NA	M
Nahlen Bose 2023 [48]	−	−	~+	−	NA	NA	NA	M
Lee/Reigel 2022 USA [49]	~	~	~	+	NA	NA	NA	H
Villero-Jimenez 2022 Spain [50] L-SP-ENG	NA	NA	+	+	NA	NA	NA	H
Ghizzardi 2022 [51]	NA	NA	−	+	NA	NA	NA	M
Suksatan 2022 Thailand [52]	NA	+	NA	NA	NA	NA	+	H
Meng 2021 China [53]	NA	NA	NA	+	NA	NA	NA	M
Tinoco 2021 Brazil [54]	NA	NA	NA	~	NA	NA	NA	M
Aghajanloo 2021 Iran [55]	NA	NA	NA	~	NA	NA	NA	L
Cañon-Montañez 2021 Colombia [56]	NA	+	~	~	NA	+	+	M
Anderson 2021 UK [57]	+	+	+	+	+	+	+	L
Zhao 2021 China [58]	NA	+	+	+	NA	NA	NA	M
Poudel 2020 USA [59]	NA	+	+	+	NA	NA	NA	M
Świątoniowska-Lonc 2020 [60]	NA	NA	−	+	NA	NA	NA	L
Peng 2019 [61]	NA	NA	+	−	NA	NA	NA	L
Parajuli 2019 Australia [62]	−	+	~	−	NA	NA	−	H
Shanbhag 2018 Canada [63]	~ +	~ +	~ +	~ +	~ +	~ +	~ +	L
Sterling 2018 USA [64]	~	~	~	~	~	~	~	L
Jiang 2018 Taiwan [65]	NA	NA	+	~ +	NA	NA	NA	M
Jonkman 2016 Holland [66]	~ +	+	+	NA	NA	NA	NA	H
Ruppar 2016 USA [67]	+	+	+	+	NA	NA	NA	H
Jonkman 2016 Holland [68]	+	+	+	~ +	NA	+	NA	H
Srisuk 2016 Thailand [69]	NA	NA	+	+	NA	NA	NA	L
Ha Dinh 2016 Vietnam [70]	NA	NA	+	+	NA	NA	NA	L
Inglis 2015 Australia [71]	+	+	+	+	NA	+	+	H
Ruppar 2015 [72]	NA	NA	NA	~	NA	NA	NA	H
Casimir 2014 [73]	−	+	+	+	NA	NA	NA	M
Wakefield 2013 USA [74]	NA	+	+	+	NA	NA	+	H
Barnason 2012 USA [75]	NA	NA	NA	+	NA	NA	NA	M
Boyde 2011 USA [76]	~	~	~	+	NA	NA	NA	M
Dickson 2011 USA [77]	NA	NA	NA	+	NA	NA	NA	H
Yehle 2010 USA [78]	NA	NA	NA	+	NA	NA	NA	H
Ditewig 2010 Holland [79]	~ +	~ +	~ +	~ +	NA	NA	NA	H
Boren 2009 USA [80]	+	NA	+	+	NA	NA	+	H
Jovicic 2006 Canada [81]	−	+	~ −	~ −	NA	NA	+	H
McAlister 2004 Canada [82]	−	+	~ +	~ +	NA	NA	+	H

**Abbreviations:** H—high; L—low; M—moderate; NA—not available; VL—very low; + positive; − negative; ~—equivocal.

**Table 3 biomedicines-13-01955-t003:** AMSTAR grading of articles and certainty of evidence.

Author (Year); Country	1	2	3	4	5	6	7	8	9	10	11	12	13	14	15	16	Total
Zhao et al., 2024 China [23]	Y	PY	Y	PY	Y	Y	N	PY	Y	N	N-MA	N-MA	N	N	N-MA	Y	L
Chen et al., 2023 Taiwan [24]	Y	PY	Y	PY	Y	Y	N	Y	Y	N	Y RCT	Y	Y	Y	Y	Y	L
Li 2023 China [25]	Y	PY	Y	PY	Y	Y	N	Y	Y RCT	N	Y RCT	Y	Y	Y	N	Y	L
Yang Mly 2023 [26]	Y	PY	Y	PY	Y	Y	N	Y	Y RCT	N	Y RCT	Y	Y	Y	Y	Y	L
Hafkamp 2022 Holland [27]	Y	PY	Y	Y	Y	Y	Y	Y	Y RCT	Y	Y RCT	Y	Y	Y	Y	Y	H
Hsu 2022 Taiwan [28]	Y	PY	Y	PY	Y	Y	Y	PY	PY	Y	Y	PY	PY	PY	N	Y	L
Toback 2017 Canada [29]	Y	PY	Y	PY	Y	Y	N	Y	Y RCT	N	Y RCT	Y	Y	Y	Y	Y	L
Taylor 2005 UK [30]	Y	PY	Y	Y	Y	Y	PY	PY	Y RCT	Y	Y RCT	Y	Y	Y	Y	Y	H
Roccaforte 2005 [31]	Y	PY	Y	PY	Y	Y	PY	PY	Y RCT	Y	Y RCT	Y	Y	Y	Y	Y	H
Gonseth 2004 [32]	Y	PY	Y	Y	Y	Y	Y	Y	Y RCT	Y	Y RCT	Y	Y	Y	Y	Y	H
Huang 2023 [33]	Y	PY	Y	PY	Y	Y	Y	Y	Y	Y	Y	Y	Y	Y	Y	Y	M
Nwosu 2023 [34]	Y	PY	Y	PY	Y	Y	Y	PY	PY	Y	Y	PY	PY	PY	N	Y	L
Checa 2022 [35]	Y	Y	Y	Y	Y	Y	Y	Y	Y RCT	Y	Y RCT	Y	Y	Y	Y	Y	H
Huang 2022 [36]	Y	PY	Y	PY	Y	Y	Y	Y	Y RCT	N	Y RCT	Y	Y	Y	Y	Y	M
Ceu 2022 [37]	Y	PY	Y	PY	Y	Y	N	Y	N BOTH	N	N BOTH	N-MA	N	N	N-MA	N	VL
Imanuel Tonapa 2022 [38]	Y	PY	Y	PY	Y	Y	PY	Y	PY RCT	N	Y RCT	N	Y	Y	Y	Y	M
Son 2020 [39]	Y	PY	Y	PY	Y	Y	PY	Y	Y RCT	Y	Y RCT	Y	Y	Y	N	Y	M
Walsh 2017 Abs Conf [40]	Y	PY	Y	PY	Y	N	N	PY	N BOTH	N	N-MA	N-MA	N	N	N-MA	N	L
Alnomasy 2023 [41]	Y	Y	Y	PY	Y	Y	Y	Y	Y	Y	Y	Y	Y	Y	Y	Y	M
Mhanna 2023 [42]	Y	PY	Y	Y	Y	Y	PY	PY	Y RCT	Y	Y RCT	Y	Y	Y	Y	Y	H
Olano-Lizarraga 2023 [43]	Y	PY	Y	PY	Y	Y	N	Y	Y RCT	N	Y RCT	Y	Y	Y	N	Y	M
Nso 2023 [44]	Y	PY	Y	PY	Y	Y	N	Y	Y RCT	N	Y RCT	Y	Y	Y	Y	Y	L
Balata 2023 [45]	Y	Y	Y	PY	Y	Y	N	Y	Y RCT	N	Y RCT	Y	Y	Y	Y	Y	M
Koikai 2023 [46]	Y	PY	Y	PY	Y	Y	N	Y	Y BOTH	N	N	N-MA	N	N	N	Y	M
Feng 2023 [47]	Y	PY	Y	PY	Y	Y	Y	Y	Y	Y	Y	Y	Y	Y	Y	Y	M
Nahlen Bose 2023 [48]	Y	PY	Y	PY	Y	N	Y	Y	Y RCT	Y	Y RCT	Y	Y	Y	Y	Y	M
Lee/Reigel 2022 USA [49]	Y	PY	Y	PY	Y	Y	N	PY	Y RCT	N	Y RCT	Y	Y	Y	Y	Y	H
Villero-Jimenez 2022 Spain [50] L-SP-ENG	Y	PY	Y	PY	Y	Y	Y	Y	Y	Y	Y	Y	Y	Y	Y	Y	M
Ghizzardi 2022 [51]	Y	PY	Y	Y	Y	Y	Y	Y	Y RCT	Y	Y RCT	Y	Y	Y	Y	Y	H
Suksatan 2022 Thailand [52]	Y	PY	Y	Y	Y	Y	Y	Y	Y BOTH	Y	Y RCT	Y	Y	Y	Y	Y	M
Meng 2021 China [53]	Y	PY	Y	PY	Y	Y	Y	PY	PY RCT	Y	Y RCT	N	Y	Y	Y	Y	M
Tinoco 2021 Brazil [54]	Y	PY	Y	PY	Y	Y	Y	PY	PY RCT	N	Y RCT	Y	Y	Y	Y	Y	L
Aghajanloo 2021 Iran [55]	Y	PY	Y	PY	Y	Y	Y	PY	Y NRSI	N	Y BOTH	Y	Y	Y	Y	Y	M
Cañon-Montañez 2021 Colombia [56]	Y	Y	Y	Y	Y	Y	Y	Y	PY RCT	Y	Y RCT	Y	Y	Y	Y	Y	M
Anderson 2021 UK [57]	Y	PY		PY	Y	Y	PY	Y	Y BOTH	N	N BOTH	N MA	Y	Y	Y	Y	L
Zhao 2021 China [58]	Y	PY	Y	PY	Y	Y	N	Y	PY RCT	N	Y RCT	N	Y	Y	Y	Y	M
Poudel 2020 USA [59]	Y	Y	Y	Y	Y	Y	Y	Y	PY RCT	Y	Y RCT	Y	Y	Y	Y	Y	M
Świątoniowska-Lonc 2020 [60]	Y	PY	Y	PY	Y	Y	PY	Y	PY RCT	Y	Y RCT	Y	Y	Y	N-MA	Y	VL
Peng 2019 [61]	Y	PY	Y	PY	Y	Y	PY	PY	Y RCT	Y	Y RCT	Y	Y	Y	N	Y	L
Parajuli 2019 Australia [62]	Y	PY	Y	Y	Y	Y	Y	Y	Y RCT	Y	Y RCT	Y	Y	Y	Y	Y	H
Shanbhag 2018 Canada [63]	Y	Y	Y	PY	Y	Y	Y	Y	Y BOTH	N	N-MA	N-MA	Y	Y	N-MA	Y	L
Sterling 2018 USA [64]	Y	Y	Y	PY	Y	Y	PY	Y	Y NRSI	Y	N-MA	N-MA	Y	Y	N	Y	L
Jiang 2018 Taiwan [65]	Y	PY	Y	PY	Y	Y	PY	PY	Y RCT	Y	Y RCT	Y	Y	Y	Y	Y	M
Jonkman 2016 Holland [66]	Y	PY	Y	PY	Y	Y	PY	Y	Y RCT	Y	Y RCT	Y	Y	Y	Y	Y	H
Ruppar 2016 USA [67]	Y	PY	Y	PY	Y	Y	N	Y	Y RCT	N	Y RCT	Y	Y	Y	Y	Y	H
Jonkman 2016 Holland [68]	Y	PY	Y	PY	Y	Y	PY	Y	Y RCT	Y	Y RCT	Y	Y	Y	Y	Y	H
Srisuk 2016 Thailand [69]	Y	PY	Y	PY	Y	Y	PY	Y	Y RCT	Y	N-MA	N-MA	Y	Y	N-MA	Y	L
Ha Dinh 2016 Vietnam [70]	Y	PY	Y	PY	Y	Y	Y	Y	Y BOTH	N	N MA	N MA	Y	Y	N-MA	Y	L
Inglis 2015 Australia [71]	Y	Y	Y	PY	Y	Y	PY	PY	Y RCT		Y RCT	Y	Y	Y	Y	Y	H
Ruppar 2015 [72]	Y	Y	Y	PY	Y	Y	PY	Y	Y RCT	Y	Y RCT	Y	Y	Y	Y	Y	H
Casimir 2014 [73]	Y	PY	Y	Y	Y	Y	PY	PY	Y RCT	N	N-MA	N-MA	Y	N	N-MA	Y	M
Wakefield 2013 USA [74]	Y	PY	Y	Y	Y	Y	Y	Y	Y RCT	Y	Y RCT	Y	Y	Y	Y	Y	H
Barnason 2012 USA [75]	Y	Y	Y	Y	Y	Y	N	Y	Y RCT/PY NRSI	Y	Y BOTH	Y	Y	N	N	Y	M
Boyde 2011 USA [76]	Y	PY	Y	PY	Y	Y	Y	Y	Y	Y	Y	Y	Y	Y	Y	Y	M
Dickson 2011 USA [77]	Y	PY	Y	Y	Y	Y	Y	Y	Y RCT	Y	Y RCT	Y	Y	Y	Y	Y	H
Yehle 2010 USA [78]	Y	PY	Y	Y	Y	Y	Y	Y	Y RCT	Y	Y RCT	Y	Y	Y	Y	Y	H
Ditewig 2010 Holland [79]	Y	PY	Y	PY	Y	Y	PY	Y	Y RCT	N	N-MA	N-MA	Y	Y	N-MA	Y	H
Boren 2009 USA [80]	Y	PY	Y	Y	Y	Y	Y	Y	Y RCT	Y	Y RCT	Y	Y	Y	Y	Y	H
Jovicic 2006 Canada [81]	Y	PY	Y	Y	Y	Y	PY	Y	Y RCT	Y	Y RCT	Y	Y	Y	Y	Y	H
McAlister 2004 Canada [82]	Y	PY	Y	PY	Y	Y	PY	Y	Y RCT	Y	Y RCT	Y	Y	Y	Y	Y	H

**Abbreviations:** AMSTRAR domain responses: N—not answered; N-MA no meta-analysis conducted; PY—partially answered; PY RCT (include only RCT); Y—yes fully answered; Y Both—(RCT and NSRI); Y RCT (include only RCTs); AMSTAR Grading: high certainty (H): we are very confident that the true effect lies close to that of the estimate of the effect; moderate certainty (M): we are moderately confident in the effect estimate: the true effect is likely to be close to the estimate of the effect, but there is a possibility that it is substantially different; low certainty (L): our confidence in the effect estimate is limited: the true effect may be substantially different from the estimate of the effect; very low certainty (VL): we have very little confidence in the effect estimate: the true effect is likely to be substantially different from the estimate of effect.

**Table 4 biomedicines-13-01955-t004:** High-quality systematic reviews, study characteristics, and findings.

(Author/Year/Country)	Study Aim/Background	Study Intervention	MA + Summary	Outcomes
Hafkamp 2022 Holland [27]	A: umbrella systematic review and meta-analyses on effectiveness of interventions in reducing HF-related (re)hospitalization	DF: pharmaceutical; device; rehabilitation multidisciplinary—variable haemodynamic, m-health, nurse-led, STS, CR. DCP	Multidisciplinary 10/23 studies (RR: 0.79, 95% CI: 0.73|0.85); CDSM (12/33 studies; RR: 0.86, 95% CI: 0.81|0.92). Limited evidence multidisciplinary clinic or SM reduces HFH Different levels of evidence regarding the effectiveness of several interventions in reducing HFH	HFH
Taylor 2005 UK [30]	A: effectiveness of disease management interventions for patients with CHF SF: inpatient, outpatient, or community based (clinical service intervention: multidisciplinary models; case management models; clinic models) EG: IP: patient; IC: care package (enhanced or novel)	DF: telephone follow-up DC: education; SM; weight-monitoring; sodium diet advice/restriction; exercise recommendation; medication review; social and psychological support	Case management interventions non-significant ↓ ACM Weak evidence that case management interventions ↓ HFH (OR 0.86, 95% CI 0.67 to 1.10, *p* = 0.23)	MACE HRQoL Cost analyses (total hospital bed days)
Roccaforte 2005 Spain [31]	A: re-evaluate the effectiveness of HFDMP on MACE and outcomes. SF: in hospital/post discharge EG: IP: patient, carer; IC: primarily education based excluded	DF: education, discharge plan, pre-planned outpatient clinic visits, home visits, tele, education session, counselling, therapy, close follow-up. Therapy optimisation, status monitoring DC: multidisciplinary approach (cardiologist, physician); case management (nurse, pharmacist, case manager)	↓ ACM, ACH—OR = 0.80 (CI 0.69–0.93, *p* = 0.003); OR = 0.58 (CI 0.50–0.67, *p* < 0.00001); OR = 0.76 (CI 0.69–0.94, *p* < 0.00001)	MACE
Gonseth 2004 Spain [32]	A: evaluate DMPs reducing hospital re-admissions among elderly SF: In hospital or post discharge; Home or outpatient visits	DF: nurse, dDischarge planning, care co-ord DC: education, counselling, and monitoring to enhance self-control mechanisms, timely medical visits, diet, and drug therapy compliance. Individualised and comprehensive patient and family HF education, follow-up and surveillance, promotion of optimal HF medications and medication doses	HFH 30% (RR 0.70; CI 95% 0.62–0.79), ACH by 12% (RR 0.88, 95% CI: 0.79–0.97) No substantial variation when only DMPs with home visits, outpatient visits to a clinic, or patient follow-up longer than 6 months were included	MACE
Checa 2022 Spain [35]	A: effect of nurse-led case management models on an advanced HF (NYHA III, IV) SF: intensive vs. basic telemedicine; home-visit interventions were intensive programmes, and basic were clinical consultations and phone calls	DF: Hosp, Comm Intensive vs. Basic DC: telemedicine, home-visit	RR 0.78, 95% CI 0.53 to 1.15; pt = 1345; studies = 6; I2 = 47% HFH (HR 0.79, 95% CI 0.68 to 0.91; pt = 1989; n = 8; I2 = 0%); ACH (HR 0.73, 95% CI 0.60 to 0.89; pt = 1012; n = 5; I2 = 36%) SMD 0.18, 95% CI 0.05 to 0.32; pt = 1228; n = 8; I2 = 28% SMD 0.66, 95% CI −0.84–2.17; pts = 450; n = 3; I2 = 97% Cost-effective at ≤EUR 60,000/QALY	MACE HRQoL Costs associated with health resources and per QALY
Mhanna 2023 Spain [42]	A: efficacy of adjunctive CBT vs. standard care in HF patients with major depression SP: Hosp, post 1–2/week up to 32 weeks; EF—N/A EG: CBT including SC intervention	DF: 30–60 m, F2F, tele DP: nurse; other not stated	Post-interventional depression scale (SMD: −0.45, 95% CI: −0.69, −0.21; *p* < 0.01) by end of follow-up (SMD: −0.68, 95% CI: −0.87, −0.49; *p* < 0.01) QoL (SMD: −0.45, 95% CI: −0.65, −0.24; *p* < 0.01) Self-care scores (SMD: 0.17, 95% CI: −0.08, 0.42; *p* = 0.18)	Primary: depression scale (post-intervention and end of follow-up) 2. Secondary: quality of life (QoL), self-care scores, 6-MWT
Lee 2022 USA [49]	A: effectiveness of self-care interventions on relevant outcomes SP: six chronic conditions. OP, home. EF-NA EG: IR: patients, carers; IC: see @	DF: face-to-face individual/group, web, media, print, phone, tools DP: multiple see @	Hedges’ g = 0.29 (95% CI = 0.25–0.33), *p* < 0.001 ↑ HRQoL	HRQOL, SC behaviour change
Villero-Jimenez 2022 Spain [50]	Identify dyadic self-management interventions in CHF in hospital settings. Hosp SP: hospital HF. NYHA I-IV EG: IR: patient carers; IC: cognitive-attitudinal, affective-emotional and behavioural	DF: variable delivery format and strategies (cognitive-attitudinal, affective-emotional, and behavioural); providers and recipients; measurement instruments used; and effectiveness. DP: nurse	↓ D, A ↑HRQoL, Ad Most of the studies demonstrated improved outcomes, especially in depression and/or anxiety symptoms, adherence to treatment, diet and weight control. Innovative design with 3 dimensions recommended	MACE HRQoL depression, anxiety, SC, adherence, and satisfaction
Suksatan 2022 Thailand [52]	A: effect of TCI on rehospitalization before discharge from hospital to home SP: Hosp, older patients, EF-NA EG: IR: HF pt; IC: discharge planning, clinic appointments, medication reconciliation, early follow-up telephone calls, HF knowledge, and SM education. TCI components: 1 = screening; 2 = staffing; 3 = maintaining relationships; 4 = engaging patients and caregivers; 5 = assessing/managing risks and symptoms; 6 = educating/promoting SM; 7 = collaborating; 8 = promoting continuity; 9 = fostering coordination	DF: F2F, home visit, tele, mob app DP: nurses, pharmacists, and multidisciplinary teams	↓ HFH costs of care	Rehospitalization within 30 days after discharge
Parajuli 2019 Australia [62]	A: evidence for the role of the pharmacist within the multidisciplinary team for HF management to improve clinical outcomes SG: hospital, outpatient clinic, or family medical practice, or under multidisciplinary HF-specialist care; NYHA II-IV EG: IR: patient; IC: pharmacist(s) working in collaboration, at a minimum, with a physician within the intervention model	DF: medication reconciliation, discharge counselling, patient education, collaborative medication management, telephone follow-up, home medication review, self-adjustment of diuretic DP: pharmacist, physician (± MDT)	↓ HFH [(OR 0.72 [95% (CI)0.55–0.93], *p* = 0.01, I2 = 39%}; ↓ ACH [OR 0.76, 95% CI (0.60–0.96), *p* = 0.02, I2 = 52%]; no effect on HFM; no effect on ACM ↑ Medication adherence; HF knowledge (*p* < 0.05), No significant improvements in healthcare costs and SM	MACE, Medication adherence (compliance), HF knowledge, health-care costs, and self-care
Jonkman 2016 Holland [66]	A: characteristics of SM interventions effective in influencing HRQol, mortality, and hospitalizations SG: Hosp, Op, Ab; EF: 39.2 EG: IR: patient, family/carer/partner; IC: SMI	DF: F2F, Tele, TM, EMep, HV DP: N, Phy, Pha, HW	↓ ACM (HR 0.99, 95% [CI] 0.97–0.999), ↓HFH (HR 0.98, 95% CI 0.96–0.99), and HFH at 6 months (risk ratio 0.96, 95% CI 0.92–0.995) Only longer programme duration improved some outcomes	MACE HRQoL
Ruppar 2016 USA [67]	A: interventions to improve medication adherence SG: Hosp, Op, Ab; EG: IR: patient, carers; IC: medication education and disease education; 11 SC	DF: F2F, Tele, TM, txt, web, Media, PM DP: MD, nurses, phar, phy, diet, SW, CM, HCW	↓ ACM (RR, 0.89;95% CI, 0.81, 0.99), ↓ HFH (OR, 0.79; 95% CI, 0.71, 0.89)	MACE
Jonkman 2016 Holland [68]	A: characteristics of SM interventions effective in influencing MACE and HRQol SG: Hosp, Op, Ab; EF: 39.2 EG: IR: patient, family/carer/partner; IC: SMI	DF: F2F, Tele, TM, EMep, HV DP: N, Phy, Pha, HW	↓ Combined endpoint HFH or ACM (HR, 0.80; 95% [CI], 0.71–0.89), time to HFH (HR, 0.80; 95% CI, 0.69–0.92) ↑ 12-month HFrQOL (SMD, 0.15; 95% CI, 0.00–0.30) HFH days <65 yo (mean, 0.70 vs. 5.35 days; *p* = 0.03) ACM with moderate/severe depression, (HR, 1.39; 95% CI, 1.06–1.83, *p* = 0.01)	MACE HRQoL
Inglis 2015 Australia [71]	A: HF management via structured telesupport SG: Hosp, Op, Am EG: IR: patient IC: disease management	DF: all options DP: all options	↓ ACM (RR 0.87, 95% CI 0.77 to 0.98; pts = 9222; studies = 22; IO = 0%, GRADE: moderate-quality evidence); HFH (RR 0.85, 95% CI 0.77 to 0.93; pts = 7030; studies = 16; IO = 27%, GRADE: moderate-quality evidence ACM, HFH	MACE
Ruppar 2015 USA [72]	A: quantify the effect of interventions to improve adherence to HF medications SG: post discharge, Op, Ab EG: IR: patients; IC: medication, disease education EG: IR: patients; IC: medication, disease education	DF: verbal (F2F, Tele) and written/electronic instructions DP: pharmacist, nurse	Effect sizes larger for studies conducted in Europe or Asia versus North America Modest ↑ SC and Adherence	Adherence (SC behaviour)
Wakefield 2013 USA [74]	A: quantify individual interventions used in multicomponent outpatient HF management programme SG: Op, Ab EG: IR: patient; IC: patient education, symptom monitoring, medication adherence strategies	DF: F2F, tele, devices, interactive videophone, scales, organiser DP: nurse	↓ HFH (ES = 0.157, *p* < 0.001); ~ MACE, SC ↓ HFrQOL (ES = 0.231, *p* = 0.007) ↓ Cost (ES = 0.17, *p* = 0.008)	Readmissions
Dickson 2011 USA [77]	A: explore how comorbidity influences HF self-care SG: NA EG: IR: patients; IC: HF education	DF: F2F, tele, group, nixed DP: NA	↓ SC with HF outcomes as patients with multiple chronic conditions are vulnerable to poor SC	SC knowledge and behaviour
Yehle 2010 USA [78]	A: how to structure educational interventions for HF patients to improve self-efficacy and SM behaviours SG: discharge, Op, Ab EG: IR: patients; IC: SM and HF education	DF: F2F, tele, group, media nixed DP: nurses, pharmacists, health educators, peer mentors	Both short and long-term interventions can improve self-efficacy, although exact ingredients are not known	SC knowledge
Ditewig 2010 Holland [79]	A: effectiveness of self-management interventions compared to usual care SG: Hosp, Op, Ab; EG: IR: patient; IC: structured HF, SC education	DF: Tele, Media, Video, pEM DP: Nurse, Phar, CM, MDT	↓ MACE ↑HRQoL (equivocal overall significance due to shortfalls in study quality)	MACE SC behaviours HRQoL
Boren 2009 USA [80]	A: identify educational content and techniques that lead to successful patient SM and improved outcome SG: inpatient, acute discharge, Op. Ab EG: IR: patient and carer; IC: structured education knowledge, social interaction and support, fluid management, diet and activity	DF: F2F, written format, health organisers, audio, video DP: nurses, pharmacists, dieticians, health educators, physician	Anecdotal improvement in MACE, HRQoL in at least one included study (no quantification provided)	SC behaviours HRQoL
Jovicic 2006 Canada [81]	A: effectiveness of SM interventions SG: Hosp, Op, Ab. EG: IR: patient, family IC: structured education	DF: F2F, Tele, SMS, Media DP: Nurse, Physician	↓ ACH (OR 0.59; 95% (CI) 0.44 to 0.80, *p* = 0.001), HFH (OR 0.44; 95% CI 0.27 to 0.71, *p* = 0.001). ACM NS (OR = 0.93; 95% CI 0.57 to 1.51, *p* = 0.76) ↑ Adherence to prescribed medical advice; HRQoL was NS Cost savings from USD 1300 to USD 7515 per patient per year	MACE HRQoL
McAlister 2004 Canada [82]	A: multidisciplinary strategies improve outcomes for heart failure SG: Hosp, Op, Ab; EG: IR: patient: IC: structured patient education	DF: Tele, TM, Media DP: MDT, Nurse, Phar, CM	MDT clinic ↓ ACM (RR 0.75, 95% [CI] 0.59 to 0.96), HFH (RR 0.74, 95% CI 0.63 to 0.87), ACH (RR 0.81, 95% CI 0.71 to 0.92) SM Programmes ↓ HFHs (RR 0.66, 95% CI 0.52 to 0.83), ACH (RR 0.73, 95% CI 0.57 to 0.93); no effect on ACM (RR 1.14, 95% CI 0.67 to 1.94) Telephone and SMS HFH (RR 0.75, 95% CI 0.57 to 0.99); not ACM (RR 0.91, 95% CI 0.67 to 1.29) or ACH (RR 0.98, 95% CI 0.80 to 1.20). Cost saving 15/18 trials	MACE HRQoL

Care Models: MD = multidisciplinary or co-led; N—nurse-led; SM—patient self-management. @ Domain Delivery: F2F—face-to-face group/individual (education, advice, and/or instruction; discussion; asking and answering questions; behavioural counselling); skills training; Tel—telephone [calls; txt—texting (e.g., using short message service or app)]; self-monitoring tools [written log or diary, digital log, or diary, wearing or using a digital device (e.g., pedometer, glucometer, or smartphone), unspecified]; web-based (E-mail, online course with feedback, E-consultation with a healthcare professional, tele-monitoring by software programme, tele-monitoring by healthcare provider, online community for peers; audio/visual/online materials [(1-way, without interaction) digital written materials, video, audio, online programme without feedback]; printed written materials (1-way, without interaction). IR—intervention recipient (patient; carer; family; partner); IC/D—intervention content/domain (SMI—self-management interventions (SMIs): Standardized Training, Multidisciplinary Team, Peer Contact, Keeping Logs, Goal-Setting Skills, Problem-Solving Skills, Seeking Support Ref. [38]. Delivery Personnel: CM—case manager; Diet—dietician; HCW—healthcare worker; MD—multidisciplinary; N—nurse; Pha—pharmacist; Phy—physician; Res—researcher; SW—social worker; Dur—duration; DF—delivery format; DP—delivery personnel. **Abbreviations:** ↓—reduce; ↑—increase; ~—equivocal; 6MWT—six-minute walk test; A—aim; Ab—ambulatory; CDSM—chronic disease self-management; CI—confidence interval; CR—cardiac rehabilitation; DC— delivery content; DF—delivery format; DP—delivery method/person; EG—experimental group; F2F—face to face; HRQoL—health-related quality of life; IC—intervention content; IR—intervention recipient; m—minutes; M—media; MA—meta-analysis; mob app—mobile phone application; MACE—major adverse cardiovascular outcomes [ACM—all-cause mortality, ACH—all-cause hospitalisation, HFH—heart failure hospitalisation, HFM—heart failure mortality]; MDT—multidisciplinary team; NA—not available; NS—not significant; OR—odds ratio; RR—relative risk; SC—service content ; SG—service ground [Ab—ambulatory; Hosp—hospital; Op—outpatient]; SM—self-management; SMS—structured messaging support; Tele—telephone]; TCI—transition care interventions. Adapted from Table A1, Ref. [10].

**Table 5 biomedicines-13-01955-t005:** (**a**) Summary of self-management (SM) behaviour outcomes across included studies. (**b**) Study quality subgroup breakdown.

(**a**)		
**Outcome**	**Number of Studies**	**References**
Significant improvement	25	[23,25,26,28,29,36,38,42,43,44,46,47,49,50,51,53,57,58,59,60,67,69,70,71,73,74,75,76,77,78,80]
Positive trend (not significant)	4	[63,65,68,79]
Equivocal findings	10	[35,37,40,41,54,55,56,64,67,81]
Negative findings	4	[39,45,48,61]
Not reported	9	[24,27,30,31,32,33,34,52,66]
(**b**)		
**Study Quality**	**Outcome Type**	**References**
**High-quality studies**	Significant improvement	[42,49,50,67,68,72,74,77,78,80]
	Equivocal	[35,68,72,79,81,82]
	Negative	[62]
	Not reported	[27,30,31,32,52,66]
**Moderate-quality studies**	Significant improvement	[33,36,38,43,46,47,51,53,58,59,73,75,76]
	Equivocal	[41,54,56,65]
	Negative	[39,45,48]
	Not reported	[33]

**Table 6 biomedicines-13-01955-t006:** (**a**) Summary of hospital readmission outcomes across included studies. (**b**) Study quality subgroup breakdown.

(**a**)		
**Outcome**	**Number of Studies**	**References**
Significant reduction	25	[23,24,27,29,31,32,35,39,41,43,46,47,52,56,57,58,59,62,66,67,68,71,73,74,81,82]
Positive trend (not significant)	3	[26,63,79]
Equivocal findings	7	[30,33,34,37,49,64,76]
Negative findings	3	[28,44,48]
Not reported	21	[25,33,38,42,45,50,51,53,54,55,60,61,65,69,70,72,75,77,78,80]
(**b**)		
**Study Quality**	**Outcome Type**	**References**
High-quality studies	Significant reduction	[27,31,32,35,62,66,67,68,71,74,81,82]
	Equivocal	[30,49,80]
	Negative	None reported
	Not reported	[42,50,67,77,78,80]
Moderate-quality	Significant reduction	[39,41,43,46,47,56,58,59,73]
	Equivocal	[33,76]
	Negative	[48]
	Not reported	[36,38,45,51,53,54,65,75]

**Table 7 biomedicines-13-01955-t007:** Meta-summary of findings across key outcomes.

Outcome	Significant Improvement	Positive Trend	Equivocal	Negative	Not Reported
HRQoL	23 studies [23,24,25,29,33,34,35,37,38,43,44,45,46,47,50,57,58,59,65,66,67,68,69,70,71,72,73,75,80]	7 studies [32,48,61,62,63,79,82]	9 studies [31,40,42,49,56,60,64,76,81]	4 studies [28,39,51,60]	14 studies [26,27,30,36,41,52,53,54,55,73,75,77,78]
Self-Management	25 studies [23,25,26,28,29,36,38,42,43,44,46,47,49,50,51,53,57,58,59,60,67,69,70,71,73,74,75,76,77,78,80]	4 studies [63,65,68,80]	10 studies [35,37,40,41,54,56,64,67,81]	4 studies [39,45,48,61]	9 studies [23,27,30,31,32,33,34,52,66]
Mortality	12 studies [23,24,26,31,32,39,43,46,57,67,68,80]	3 studies [63,66,79]	6 studies [30,33,34,49,64,76]	10 studies [27,28,35,44,47,48,49,62,81,82]	28 studies [25,29,36,37,38,40,41,42,45,50,51,52,53,54,55,56,58,59,60,61,65,69,70,72,74,75,77,78]
Readmissions	25 studies [23,24,27,29,31,32,35,39,41,43,46,47,52,56,57,58,59,62,66,67,68,71,73,74,81,82]	3 studies [26,63,80]	7 studies [30,33,34,37,49,64,76]	3 studies [28,44,48]	21 studies [25,33,38,42,45,50,51,53,54,55,60,61,65,69,70,72,75,77,78,80]

## Data Availability

The original contributions presented in this study are included in the article. Further inquiries can be directed to the corresponding author.

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
