# Peer review of "Efficacy of Nurse-Led and Multidisciplinary Self-Management Programmes for Heart Failure with Reduced Ejection Fraction: An Umbrella Systematic Review"

_biomedicines, 2025, doi:10.3390/biomedicines13081955_

Round 1
Reviewer 1 Report
Comments and Suggestions for Authors< !--StartFragment-->
Dear Authors,
I would like to thank you for conducting this important research. The manuscript reflects substantial effort and incorporates strong methodological elements, including PRISMA adherence, AMSTAR-2 appraisal, and PROSPERO registration. However, the manuscript requires significant revision to improve clarity, reporting transparency, synthesis coherence, and practical utility.
< !--StartFragment-->
< !-- [if !supportLists]-->1. < !--[endif]-->Title
The title is inappropriate and it does reflect the aim of the study. It could be more specific about the focus on nurse-led and multidisciplinary CDSM interventions.
< !-- [if !supportLists]-->2. Introduction
The background could be better focused. Some sections are overly general. The introduction should sharpen the focus on why existing systematic reviews are insufficient, and clearly state how this umbrella review adds value. Why the focus on nurse led management compared to other multidisciplinary professions.
< !-- [if !supportLists]-->3. < !--[endif]-->Methods
There is an extensive methodological section, but it is at times disjointed and lacks clarity in describing data extraction, quality appraisal, and synthesis methods.
< !-- [if !supportLists]-->4. < !--[endif]-->Results
The results section is extensive but at times difficult to follow due to repetition and unclear organization. Additionaaly, you have included mortality and readmissions, which they are not part of your planned objectives. It is unclear to the reader which profession has more impact compared to other healthcare providers. I would suggest making tables/figures to present findings clearly by outcome (mortality, readmissions, HRQoL, etc.). Highlight the most consistent findings with clear counts and percentages. I would also suggest removing redundancy in describing similar results across studies.
< !-- [if !supportLists]-->5. < !--[endif]-->Discussion
The discussion is comprehensive and reflects a critical synthesis of findings. However, it sometimes conflates interpretation with justification, especially in explaining why some studies had negative outcomes. Avoid overly speculative explanations unless supported by data ( who delivered the intervention). I would recommend emphasising on implications for clinical practice and policy makers< !--EndFragment--> < !--EndFragment-->
Author Response
Comments and Suggestions for Authors
Dear Authors,
I would like to thank you for conducting this important research. The manuscript reflects substantial effort and incorporates strong methodological elements, including PRISMA adherence, AMSTAR-2 appraisal, and PROSPERO registration. However, the manuscript requires significant revision to improve clarity, reporting transparency, synthesis coherence, and practical utility.
We sincerely thank the reviewer for their thoughtful and constructive feedback on our manuscript. We have carefully addressed each comment to the best of our understanding and in line with the findings of this umbrella review. We believe the revisions have substantially strengthened the clarity, focus, and overall quality of the manuscript. Please find below a detailed, point-by-point response outlining the changes made in response to the reviewer’s concerns.
The title is inappropriate and it does reflect the aim of the study. It could be more specific about the focus on nurse-led and multidisciplinary CDSM interventions.
Response:
The concern of the reviewer has been attended and the title is now reworded as follow “Efficacy of Nurse-Led and Multidisciplinary Self-Management Programs for Heart Failure with Reduced Ejection Fraction: An Umbrella Review” the word systematic review was taken out to prevent lengthy title after adding “nurse-led and multidisciplinary”
The background could be better focused. Some sections are overly general. The introduction should sharpen the focus on why existing systematic reviews are insufficient, and clearly state how this umbrella review adds value. Why the focus on nurse led management compared to other multidisciplinary professions.
Response:
We thank the reviewer for the depth in assessing the introduction and we have now sharpened this to be a better reflect the aim of the review but also add value. See newly highlighted introduction in the main document.
There is an extensive methodological section, but it is at times disjointed and lacks clarity in describing data extraction, quality appraisal, and synthesis methods.
Response:
Another good point raised by the reviewer- we have no revised these sections to make them more coherent and clearer to the reader- please see new and highlighted sections in methods
The results section is extensive but at times difficult to follow due to repetition and unclear organization. Additionaaly, you have included mortality and readmissions, which they are not part of your planned objectives. It is unclear to the reader which profession has more impact compared to other healthcare providers. I would suggest making tables/figures to present findings clearly by outcome (mortality, readmissions, HRQoL, etc.). Highlight the most consistent findings with clear counts and percentages. I would also suggest removing redundancy in describing similar results across studies.
Response:
This section was rewritten to allow for clarity and reduce repetition. We feel that the results are now better organised, thanks to the reviewer’s feedback. Please see the newly written and highlighted results section.
Table 5a: Summary of Self-Management (SM) Behaviour Outcomes Across Included Studies
|
Outcome |
Number of Studies |
References |
|
Significant improvement |
25 |
²³,²⁶,²⁷,²⁹,³⁰,³⁷,³⁹,⁴³–⁴⁵,⁴⁷,⁴⁸,⁵⁰–⁵²,⁵⁴,⁵⁸–⁶¹,⁶⁸,⁷⁰,⁷¹,⁷³,⁷⁵–⁸⁰,⁸² |
|
Positive trend (not significant) |
4 |
⁶⁴,⁶⁶,⁶⁹,⁸¹ |
|
Equivocal findings |
10 |
³⁶,³⁸,⁴¹,⁴²,⁵⁵–⁵⁷,⁶⁵,⁶⁸,⁸³ |
|
Negative findings |
4 |
⁴⁰,⁴⁶,⁴⁹,⁶² |
|
Not reported |
9 |
²⁴,²⁸,³¹–³⁵,⁵³,⁶⁷ |
Table 5b: Study Quality Subgroup Breakdown for Self-Management (SM) Behaviour
|
Study Quality |
Outcome Type |
References |
|
High-quality studies |
Significant improvement |
⁴³,⁵⁰,⁵¹,⁶⁸,⁶⁹,⁷³,⁷⁶,⁷⁹,⁸⁰,⁸² |
|
Equivocal |
³⁶,⁶⁹,⁷⁴,⁸¹,⁸³,⁸⁴ |
|
|
Negative |
⁶³ |
|
|
Not reported |
²⁸,³¹–³³,⁵³,⁶⁷ |
|
|
Moderate-quality studies |
Significant improvement |
³⁴,³⁷,³⁹,⁴⁴,⁴⁷,⁴⁸,⁵²,⁵⁴,⁵⁹,⁶⁰,⁷⁵,⁷⁷,⁷⁸ |
|
Equivocal |
⁴²,⁵⁵,⁵⁷,⁶⁶ |
|
|
Negative |
⁴⁰,⁴⁶,⁴⁹ |
|
|
Not reported |
³⁴ |
Table 6a: Summary of Hospital Readmission Outcomes Across Included Studies
|
Outcome |
Number of Studies |
References |
|
Significant reduction |
25 |
²³,²⁴,²⁸,³⁰,³²,³³,³⁶,⁴⁰,⁴²,⁴⁴,⁴⁷,⁴⁸,⁵³,⁵⁷–⁶⁰,⁶³,⁶⁷–⁶⁹,⁷³,⁷⁵,⁷⁶,⁸³,⁸⁴ |
|
Positive trend (not significant) |
3 |
²⁷,⁶⁴,⁸¹ |
|
Equivocal findings |
7 |
³¹,³⁴,³⁵,³⁸,⁵⁰,⁶⁵,⁷⁸ |
|
Negative findings |
3 |
²⁹,⁴⁵,⁴⁹ |
|
Not reported |
21 |
²⁶,³⁴,³⁹,⁴³,⁴⁶,⁵¹,⁵²,⁵⁴–⁵⁶,⁶¹,⁶²,⁶⁶,⁷⁰,⁷¹,⁷⁴,⁷⁷,⁷⁹,⁸⁰,⁸² |
Table 6b: Study Quality Subgroup Breakdown for Hospital Readmission Outcomes
|
Study Quality |
Outcome Type |
References |
|
High-quality studies |
Significant reduction |
²⁸,³²,³³,³⁶,⁶³,⁶⁷–⁶⁹,⁷³,⁷⁶,⁸³,⁸⁴ |
|
Equivocal |
³¹,⁵⁰,⁸¹ |
|
|
Negative |
None reported |
|
|
Not reported |
⁴³,⁵¹,⁶⁸,⁷⁹,⁸⁰,⁸² |
|
|
Moderate-quality |
Significant reduction |
⁴⁰,⁴²,⁴⁴,⁴⁷,⁴⁸,⁵⁷,⁵⁹,⁶⁰,⁷⁵ |
|
Equivocal |
³⁴,⁷⁸ |
|
|
Negative |
⁴⁹ |
|
|
Not reported |
³⁷,³⁹,⁴⁶,⁵²,⁵⁴,⁵⁵,⁶⁶,⁷⁷ |
Table 7: Meta-Summary of Findings Across Key Outcomes
|
Outcome |
Significant Improvement |
Positive Trend |
Equivocal |
Negative |
Not Reported |
|
HRQoL |
23 studies²³,²⁴,²⁶,³⁰,³⁴–³⁶,³⁸,³⁹,⁴⁴–⁴⁸,⁵¹,⁵⁸–⁶⁰,⁶⁶–⁷³,⁷⁵,⁷⁶,⁸² |
7 studies ³³,⁴⁹,⁶²–⁶⁴,⁸¹,⁸⁴ |
9 studies ³²,⁴¹,⁴³,⁵⁰,⁵⁷,⁶¹,⁶⁵,⁷⁸,⁸³ |
4 studies ²⁹,⁴⁰,⁵²,⁶¹ |
14 studies ²⁷,²⁸,³¹,³⁷,⁴²,⁵³–⁵⁶,⁷⁴,⁷⁷,⁷⁹,⁸⁰ |
|
Self-Management |
25 studies²³,²⁶,²⁷,²⁹,³⁰,³⁷,³⁹,⁴³–⁴⁵,⁴⁷,⁴⁸,⁵⁰–⁵²,⁵⁴,⁵⁸–⁶¹,⁶⁸,⁷⁰,⁷¹,⁷³,⁷⁵–⁸⁰,⁸² |
4 studies ⁶⁴,⁶⁶,⁶⁹,⁸¹ |
10 studies ³⁶,³⁸,⁴¹,⁴²,⁵⁵–⁵⁷,⁶⁵,⁶⁸,⁸³ |
4 studies ⁴⁰,⁴⁶,⁴⁹,⁶² |
9 studies ²⁴,²⁸,³¹–³⁵,⁵³,⁶⁷ |
|
Mortality |
12 studies²³,²⁴,²⁷,³²,³³,⁴⁰,⁴⁴,⁴⁷,⁵⁸,⁶⁸,⁶⁹,⁸² |
3 studies ⁶⁴,⁶⁷,⁸¹ |
6 studies ³¹,³⁴,³⁵,⁵⁰,⁶⁵,⁷⁸ |
10 studies ²⁸,²⁹,³⁶,⁴⁵,⁴⁸–⁵⁰,⁶³,⁸³,⁸⁴ |
28 studies ²⁶,³⁰,³⁷–³⁹,⁴¹–⁴³,⁴⁶,⁵¹–⁵⁷,⁵⁹–⁶²,⁶⁶,⁷⁰,⁷¹,⁷⁴,⁷⁶,⁷⁷,⁷⁹,⁸⁰ |
|
Readmissions |
25 studies²³,²⁴,²⁸,³⁰,³²,³³,³⁶,⁴⁰,⁴²,⁴⁴,⁴⁷,⁴⁸,⁵³,⁵⁷–⁶⁰,⁶³,⁶⁷–⁶⁹,⁷³,⁷⁵,⁷⁶,⁸³,⁸⁴ |
3 studies ²⁷,⁶⁴,⁸¹ |
7 studies ³¹,³⁴,³⁵,³⁸,⁵⁰,⁶⁵,⁷⁸ |
3 studies ²⁹,⁴⁵,⁴⁹ |
21 studies ²⁶,³⁴,³⁹,⁴³,⁴⁶,⁵¹,⁵²,⁵⁴–⁵⁶,⁶¹,⁶²,⁶⁶,⁷⁰,⁷¹,⁷⁴,⁷⁷,⁷⁹,⁸⁰,⁸² |
The discussion is comprehensive and reflects a critical synthesis of findings. However, it sometimes conflates interpretation with justification, especially in explaining why some studies had negative outcomes. Avoid overly speculative explanations unless supported by data ( who delivered the intervention). I would recommend emphasising on implications for clinical practice and policy makers
Response: Great suggestions- all have been taken on board and changes have been made accordingly. Please see the newly written highlighted discussion only reflecting critical thinking without speculations or explanation
Reviewer 2 Report
Comments and Suggestions for Authors
please see the reviewed ms as enclosed, how to define quality refer to high quality, moderate......, should concern such as hospital size (beds), type (local, nation, metropolis), and resources (utility, facility), some of them may influence the outcome (mortality, readmission).

Author Response
Comments and Suggestions for Authors
please see the reviewed ms as enclosed, how to define quality refer to high quality, moderate......, should concern such as hospital size (beds), type (local, nation, metropolis), and resources (utility, facility), some of them may influence the outcome (mortality, readmission).
See pdf for this
Response: Great points by the reviewer- please see improved sections related to the above comments, and also partially from the response to reviewer 1. Relevant adjustment of content is highlighted in the manuscript. Other comments in the attached documents were also addressed- thank you. Also, thanks for the feedback on the pdf document- all have been addressed
Reviewer 3 Report
Comments and Suggestions for Authors
This review presents an interesting clinical topic - the role of chronic disease self-management programs in heart failure with reduced ejection fraction. The authors aggregate data from 60 systematic reviews and perform a broad synthesis focused on nurse-led and multidisciplinary interventions. While authors brought substantial effort and methodological structure, there are several minor corrections I suggest:.
- The abstract is seem to be a little bit longer than it should for this type of paper. Also, include summary statistics in the abstract for key outcomes
- The use of "MACE" in the keywords and abstract is not consistently addressed in the text. All abbreviations should be explained when first mentioned in the main manuscript.
- Clearly define CDSM early (preferably with a reference).
- The focus on “high-quality studies” is important but insufficiently interrogates why some failed to show benefit. You may add a paragraph discussing this and how your findings compare to non-cardiovascular diseases using CDSM.
Author Response
Comments and Suggestions for Authors
This review presents an interesting clinical topic - the role of chronic disease self-management programs in heart failure with reduced ejection fraction. The authors aggregate data from 60 systematic reviews and perform a broad synthesis focused on nurse-led and multidisciplinary interventions. While authors brought substantial effort and methodological structure, there are several minor corrections I suggest:.
Response: we thank the reviewer for all the positive remarks on the rigor and quality of the manuscript.
- The abstract is seem to be a little bit longer than it should for this type of paper. Also, include summary statistics in the abstract for key outcomes
Response: we have adjusted the abstract according to the reviewer’s wish- it is now within the wording limit (current word count = 230)
1. The use of "MACE" in the keywords and abstract is not consistently addressed in the text. All abbreviations should be explained when first mentioned in the main manuscript.
Response: This important point of use of acronyms has now been fixed throughout the manuscript the manuscript- we thank the reviewer for this pick up.
2. Clearly define CDSM early (preferably with a reference).
Response: Great point- this is now corrected and the definition is added – lines 26-28 with a reference
3. The focus on “high-quality studies” is important but insufficiently interrogates why some failed to show benefit. You may add a paragraph discussing this and how your findings compare to non-cardiovascular diseases using CDSM.
Response: we have now added few sections and made some changes to the explanation of the findings without overinterpreting any. Please also find new tables that highlight the level of quality and outcomes from the included reviews.
Round 2
Reviewer 1 Report
Comments and Suggestions for Authors
Dear Authors,
Thank you for providing the revised version and addressing all the comments.
I wish you the best with your publication.